# Physiological Responses and Nutritional Intake during a 7-Day Treadmill Running World Record

**DOI:** 10.3390/ijerph17165962

**Published:** 2020-08-17

**Authors:** Nicolas Berger, Daniel Cooley, Michael Graham, Claire Harrison, Russ Best

**Affiliations:** 1School of Health and Life Sciences, Teesside University, Middlesbrough TS1 3BX, UK; D.Cooley@tees.ac.uk (D.C.); Michael.Graham@tees.ac.uk (M.G.); 2Newcastle Nutrition Community Team, The Newcastle Upon Tyne Hospitals NHS Foundation Trust, Newcastle NE7 7DN, UK; claire@c-harrison.co.uk; 3Centre for Sports Science & Human Performance, WINTEC, Hamilton 3200, New Zealand

**Keywords:** ultra-endurance, case-study, running, world-record, nutrition, exercise performance

## Abstract

Ultra-running comprises running events longer than a marathon (>42.2 km). The prolonged duration of ultra-running leads to decrements in most or all physiological parameters and considerable energy expenditure (EE) and energy deficits. SG, 47 years, 162.5 cm, 49 kg, VO_2max_ 4 mL/kg/min^−1^/2.37 L/min^−1^, ran continuously for 7 days on a treadmill in 3 h blocks followed by 30 min breaks and slept from 1–5 a.m. Heart rate (HR) oxygen uptake (VO_2_), rating of perceived exertion, weight, blood lactate (mmol·L^−1^), haemoglobin (g·dL), haematocrit (%) and glucose (mmol·L^−1^), and nutrition and hydration were recorded. SG ran for 17.5 h/day, covering ~120 km/day at ~7 km/h. Energy expenditure for each 24 h period was 6878 kcal/day and energy intake (EI) was 2701 kcal/day. EE was 382 kcal/h, with 66.6% from fat and 33.4% from carbohydrate oxidation. 7 day EI was 26,989 kcal and EE was 48,147 kcal, with a total energy deficit (ED) of 21,158 kcal. Average VO_2_ was 1.2 L·min^−1^/24.7 mL·kg·min^−1^, Respriatory echange ratio (RER) 0.80 ± 0.03, HR 120–125 b·min^−1^. Weight increased from 48.6 to 49.5 kg. Haemoglobin decreased from 13.7 to 11 g·dL and haematocrit decreased from 40% to 33%. SG ran 833.05 km. SG exhibits an enhanced fat metabolism through which she had a large daily ED. Her success can be attributed to a combination of physiological and psychological factors.

## 1. Introduction

Ultra-running comprises running events longer than a marathon (>42.2 km), with races contested over multiple distances (e.g., 50 km to 100 miles) or time periods (e.g., 24 h) in single-day and multi-day formats. The longest event held over km is 1000 km and for events held in miles, the longest distance is the “Self-Transcendence 3100 Mile Race” (4989 km) (www.3100.ws). Ultra-running events take place on a variety of terrains, ranging from treadmills to remote wilderness, and these variations not only affect the biomechanical, physiological, nutrition, and psychological [1,2,3,4,5] demands of ultra-running, but also the strategies employed by athletes and their support crews. Due to the prolonged duration of ultra-running, decrements in most or all “systems” is to be expected.

Examples of decrements in physiological parameters include a decrease in body mass and dehydration, loss of skeletal muscle mass, and increased total body water [6]. It is thought that an increase in body water could be the result of protein catabolism and the corresponding development of hypoproteinaemic oedema [7]. The resulting increase in plasma proteins and increase in plasma volume are suggested to be the result of increased plasma sodium concentration and increased activity of vasopressin. The retention of sodium is most likely due to increased activity of aldosterone [8,9,10]. Knechtle et al. [11] showed during a 17 day, 1200 km race that this effect is cumulative, with an increase in body water as the race progressed. In an excellent review by Knechtle and Nikolaidis [12], the physiology of ultra-running is described in detail and the interested reader is directed there for in-depth information.

Decrements in physiological parameters are likely worsened by in race nutrition scenarios. Energy expenditure is often not commensurate with energy intake in ultra-running, leading to marked decreases in body fat and muscle mass [13]. Typical daily energy deficits are substantial and can be further exacerbated as a result of gastro-intestinal (GI) distress and limited appetite [13], but also due to the inability to carry sufficient supplies in self-supported races or logistical challenges imposed by feeding station and support crew regulations.

Performance and recovery are heavily influenced by appropriate nutritional intake during training and competition [14,15], but it has been demonstrated that ultra-running competitors may not meet nutritional intake guidelines, mainly due to logistical constraints and gastro-intestinal (GI) distress [16,17]. Carbohydrate (CHO) ingestion can enhance endurance performance [1,18,19]. The recommendation for athletes who want to perform prolonged or repeated endurance activity is to ingest 60–120 g/h of CHO during activity to minimise the depletion of intramuscular and hepatic glycogen stores, which also limits exercise-induced muscle damage and shortens recovery time [20,21,22]. However, the ingestion of large quantities of CHO has also been shown to cause GI distress and it is advised that ultra-endurance athletes are fueled by a combination of different types of CHO to minimise this effect [23]. It is also possible to attenuate GI distress through nutritional training and physiological adaptation, where the athlete practises ingesting larger volumes of CHO during training [14,23]. Recommended protein intake for athletes involved in high volume or intense exercise is 1.7–2.2 g/kg/day [15,24,25]; more specifically, older athletes should ingest single doses of 40 g protein post exercise as older muscles respond more slowly and are less sensitive to protein ingestion due to higher protein synthesis thresholds [26,27,28]. The recommendations for fat intake are to consume a moderate amount, which should make up approximately 30% of their daily calorie intake [15,29].

We present data on a female multiple world record holding ultra-runner, examining haematological and physiological perturbations, as well as nutritional and recovery strategies, throughout a successful treadmill world record attempt for total distance completed in seven days on a treadmill (833.05 km). Historically, our athlete displayed very low energy intake (EI) during running events that is not commensurate with energy expenditure (EE). Despite this, her performance has not been affected in past events of very long duration and intensity. In fact, she has often won, placed very highly, or set new records, often beating men. Therefore, it was our aim to monitor closely the EE and EI during seven days of continuous exercise to quantify the large energy deficit (ED) and any concomitant decrements in performance, as well as describing haematological or physiological parameters.

## 2. Materials and Methods

### 2.1. Participant

SG is a female ultra-endurance runner with almost 40 years’ competitive experience. At the time of the event in 2011, she was 47 years old. She was not athletic as a teenager and did not start running until she was 22 years old. SG ran her first marathon in 1986 and has competed internationally for Great Britain at 100 km and 24 h, has completed more than 50,000 km of racing, running more than 400 marathons and over 150 ultra-endurance events. She holds multiple world records, the most recent being set in July 2019, where she broke the Land’s End to John O’Groats World Record (covering the whole length of Great Britain, between two extremities, in the southwest and northeast) by over 4 h (837 miles in 12 days, 11 h, 6 min, and 7 s).

This study was approved by Teesside University and the Waikato Institute of Technology (12/2011-TSWT201112URWRSG).

### 2.2. Record Attempt

The record consisted of running as far as possible (cumulative distance) on a treadmill over seven days. SG successfully completed a world record attempt, in accordance with Guinness World Record requirements. To be eligible for a record, efforts must be based upon one variable that is breakable, measurable, standardisable, and verifiable (Guinness World Records Ltd., London, UK, 2019), with supporting evidence submitted as per Guinness World Records requirements (Guinness World Records Ltd., 2019).

The attempt took place during December 2011 in an open area of Teesside University, in the UK. SG chose the location due to access to three suitable treadmills (Quasar, H/P/Cosmos Sports, and Medical GMBH, Nussdorf, Traunstein, Germany), general facilities, and scientific testing equipment. Temperature was kept at a constant 20 °C throughout the attempt. SG ran in 3 h blocks followed by 30 min recovery periods to use the toilet, eat/drink, receive treatment (massage, ice bath for feet), and to allow for data collection. During each 3 h block, heart rate (HR), breath by breath oxygen uptake (VO_2_), and overall rating of perceived exertion (RPE) were assessed. The athlete had a complete rest from 1–5 a.m. and slept in the same room that the treadmill was located in. The accompanying research crew consisted of members of University staff who rotated in shifts throughout the record attempt.

### 2.3. Measures

#### 2.3.1. Body Composition and Haematology

Body mass was assessed before the start of the attempt and during every 30 min break (Seca 869, Birmingham, UK). Body mass (kg) was expressed as a mean weight per day. Prior to analysis, all blood samples were taken from a fingertip capillary of the ring finger of the left hand. Baseline values were collected pre-attempt and immediately after each 3 h running block and analysed for blood lactate (mmol·L^−1^; Ysi, 2300 Lactate and Glucose, Ysi UK Ltd., Hampshire, Cardiff, UK), haemoglobin (g·dl; B-Hemoglobin, HemoCue Ltd., Sheffield, UK), and glucose (mmol·L^−1^; Reflotron Plus (Reflotron, Roche Diagnostics GmbH, D68298 Mannheim, Germany). Samples for haematocrit were collected in heparinised glass tubes and spun for 5 min at 14,000 rpm (HaematoSpin 1400). Spun blood was analysed using a Hawksley micro haematocrit reader (% Hct; Hawksley, UK, Lancing, Sussex) to determine the percentage of red blood cells in the whole blood.

#### 2.3.2. Nutritional Intake

Nutritional intake was recorded at time of consumption by the support crew in real time and food was weighed pre and post in cases of prepared meals. Intake was analysed for total energy (kcal) and absolute (g·h^−1^) intakes for macronutrients per day using specialist software (Nutritics v5.096, Nutritics, Dublin, Ireland). Relative intake (g·kg^−1^) and percent contribution towards the total of each macronutrient for each day was also calculated.

Initially, SG planned to eat a pre-cooked CHO and protein-rich meal every three hours during scheduled 30 min breaks to minimise the need for food intake during periods of running. The food was prepared by SG beforehand and frozen, to be microwaved and ready for consumption. Examples of dishes include fish pie (pieces of fish, white sauce, peas, and mashed potato) and pasta with bolognese sauce, mincemeat and rice, and bean stew with rice. SG reported not being able to consume such large quantities in such a short space of time, as well as the food not being prepared in sufficient time for it to cool enough for her to consume it quickly. Therefore, from the middle of day 2 onwards, she only consumed one more cooked meal, instead relying upon cold foods, such as sandwiches, cereal, fruit smoothies, fruit, energy drinks, and chocolate (Table 1), avoiding heavier foods, which SG attributed to GI distress. Apart from occasional CHO and electrolyte sports drink ingestion, SG did not rely on any specific sports nutrition products, instead consuming a lot of her CHO via fruit smoothies. SG did not supplement with anhydrous caffeine and only consumed moderate amounts via instant coffee and tea. No other supplements were taken.

#### 2.3.3. Cardio-Respiratory and Subjective Measures

Heart rate (HR) was recorded every five seconds via telemetry (PE4000, Polar Electro Oy, Kempele, Finland). Due to an equipment/software failure, we only have data for one 24 h period (Day 3) for HR. We used VO_2_ to calculate EE and as such, did not rely on HR data. The corresponding VO_2_ data show minimal fluctuations, so we can confidently extrapolate that HR was stable throughout the event. Gas exchange data was determined for 3 min at the end of each 3 h block using breath by breath online gas analysis (Zan, 600 USB, CPXnSpire Health Inc., Hertford, UK), for the derivation of VO_2_, VCO_2_, and RER. The average of each 3 min sample was calculated from the raw, unfiltered breath by breath data. From measures of VCO_2_ and VO_2_, total carbohydrate (CHO) and fat oxidation rates were calculated using the formulas of Frayn [30]:
Total fat oxidation (g/min^−1^) = 1.67 × VO_2_ − 1.67 × VCO_2_
Total CHO oxidation (g/min^−1^) = 4.55 × VCO_2_ − 3.21 × VO_2_.

Although not possible to assess protein oxidation with our methods, we must assume some contribution of protein metabolism of approximately 5–15% towards energy production [31,32], as a loss of muscle mass in multi-day ultra-running is very common. Knechtle & Kohler [33] showed that a 5 day, 338 km stage race led to decreases in skeletal muscle, but not fat mass. As such, the calculated EE values from CHO and fat alone are likely slightly overestimated.

Overall rating of perceived exertion (RPE) was assessed via the CR-10 scale [34] and was recorded at the end of each three-hour block. Arbitrary units are reported with accompanying verbal descriptors.

### 2.4. Statistical Analysis

Data are reported as means ± standard deviations, unless otherwise stated.

## 3. Results

### 3.1. Record Breaking Performance

SG successfully broke the record for seven consecutive days on a treadmill, covering 833.05 km in a total running time of 120 h 54 min 28 s, beating the previous female record holder by 182.73 km and previous male record holder by 79.33 km.

### 3.2. Laboratory Testing before the Event

SG was 162.5 cm and 49 kg and in a VO_2max_ test from October 2011, she reached a vVO_2max_ of 16 km/h and achieved a VO_2max_ of 48 mL·kg^−1^·min^−1^ (2.37 L·min^−1^). Lactate threshold occurred at 13 km·h^−1^, at a HR of 157 b·min^−1^, corresponding to approximately 80% VO_2max_.

### 3.3. Energy Intake

Energy and absolute and relative macronutrient intakes for each day of the seven-day attempt are presented in Table 2 and Table 3. During the attempt, the total energy intake was 18,918 kcal, with an average of 2702 ± 466 kcal·day^−1^, consisting of 441 ± 137 g CHO, 97 ± 33 g protein, and 64 ± 21 g fat. Of the total calories, 65% were from CHO, 14% from protein, and 21% from fat. SG consumed a total of 26.2 litres of fluid over the 7 days, averaging 3.6 ± 1 L·day^−1^. Sodium intake was 33,540 mg over the 7 days, averaging 4577 ± 967 mg·day^−1^.

### 3.4. Energy Expenditure and Oxygen Uptake during the Attempt

Calculation of energy expenditure (EE) and contribution towards EE from CHO and fats was derived from an unfiltered 3 min breath by breath VO_2_ sample during every 3 h block. The average VO_2_ for the 7 days was 1.2 ± 0.1 L·min^−1^ and 24.7 ± 3.4 mL·kg^−1^·min^−1^. Therefore, SG performed at ~50% of her VO_2max_ for the entirety of the event. Respiratory exchange ratio (RER) averaged 0.8 ± 0.03 for the 7 days, with SG oxidising 33.4% CHO and 66.6% fat on average, respectively (Figure 1).

SG expended 48,147 kcal over the 7 days (including rest periods). This equates to an EE of ~286 kcal·h^−1^ with rest periods and sleep, or 382 kcal·h^−1^ for active periods only. Energy deficit (ED; calculated as EI – EE) was calculated from continuous VO_2_ files and was 29,229 kcal over the 7 days, or 4174 ± 760 kcal·day^−1^.

### 3.5. Blood Parameters

Haematology values decreased over the 7 days. Haemoglobin decreased from 13.7 g/dL on day 1 to 11.6 g/dL on day 7 (Figure 2A). There was a steady decline, with the lowest reading of 9.7 g/dL on day 5. Haematocrit followed the same trend, starting at 40% on day 1 and decreasing to 33% on day 7, with the lowest reading of 25% on day 5 (probably related to haemodilution and not a result of low red blood cell count; see discussion for explanation. Figure 3A). As expected, there were some large fluctuations in blood glucose, with a starting value of 6.28 mmol·L^−1^ on day 1. The lowest and highest readings were on day 2, with values of 2.65 mmol·L^−1^ and 9.14 mmol·L^−1^, respectively (Figure 2B). Following a high reading of 9.09 mmol·L^−1^ on day 3, blood glucose remained fairly stable between 6 and 8 mmol·L^−1^. Lactate only exceeded 3 mmol·L^−1^ on day 6 and remained between 0.9–2 mmol·L^−1^ for the majority of the attempt (Figure 4A).

### 3.6. Rating of Perceived Exertion

RPE remained mostly stable throughout the seven-day attempt (Figure 3B), with a mean rating of 4.08 ± 0.94 arbitrary units, corresponding to a perceived effort of *Somewhat Hard*, ranging from *Easy* (2) to *Hard* (6). RPE reached 6 once on day 3 and twice on day 6. At the start of day 4, at 8 a.m., RPE was recorded as 2, which coincided with a period of walking.

### 3.7. Body Mass

SG finished the event 1 kg heavier than she started; 48.5 kg day 1 vs. 49.5 kg day 7 (increase of 2.06%). Mean body weight for the attempt was 48.5 ± 0.71 kg; however, there was a trend for bodyweight to increase over the course of the seven days, as shown in Figure 4B.

## 4. Discussion

We present data from a successful world record attempt set in the UK in 2011. SG broke the previous record of 753.24 km by 79 km for running continuously on a treadmill for 7 days and set a new world record of 833.05 km. We assessed EI (via detailed food diaries; see Table 1), EE (assessed using VO_2_), fluid intake, blood lactate, haematocrit, haemoglobin, blood glucose, body mass, and overall rating of perceived exertion every 3 h over the 7 day world record attempt.

Although the research on ultra-endurance performance is growing, many of the studies are lacking important data on participants; e.g., lactate threshold, VO_2max_, etc. Therefore, we sought to provide a comprehensive overview of an athlete and detail the physiological changes during a successful ultra-endurance attempt. Previous research has shown that training volume and intensity, best marathon time, skinfold thickness, VO_2max_, and peak treadmill velocity predict ultra-running performance [3]. SG’s results from a VO_2max_ test completed shortly before the record attempt showed a relatively ordinary VO_2max_ of 48 mL·kg^−1^·min^−1^; this is not a value that would seem to predict the ability to set world records. Despite SG’s frankly modest VO_2max_ result, her continuously low VO_2_ and RER (approximately 1.2 L·min^−1^ and 0.8, respectively) during the event suggests that she utilises fat and CHO efficiently, thereby reducing the energetic cost of exercise [35].

The main finding from our investigation was that running for 7 consecutive days led to a significant daily ED which did not result in a commensurate weight loss. SG did not consume a lot of food during the 3 h running blocks, mainly eating during breaks. This led to a relatively high, but mainly stable blood glucose level for much of the event (Figure 2B). A study of energy balance in a male ultra-endurance runner during a 24-h treadmill run indicated that despite a substantial ED of ~8000 kcal, blood glucose remained within normal values undefined. It was hypothesised that as well as energy derived from CHO and fat metabolism, a substantial provision came from lactate and gluconeogenesis [36]. In the current study, the lowest blood glucose level and highest lactate level appear to coincide (Figure 2B and Figure 4A).

During multi day events, ultra-running athletes seldom consume sufficient energy to meet the estimated requirements based upon extrapolation of daily EE predictions for typical activities [13,37]. Compounding factors such as logistical constraints, sleep deprivation, and physical tolerance to foods and fluids all pose challenges for such athletes. Some events may lend themselves to pre weight gain in order to offset losses. However, this is not always feasible nor desirable in running based events, due to their weight-bearing nature, which would increase the physical strain and energetic cost of the activity [38]. The intake observed in the current case study during a 7-day treadmill run typifies the type of intake observed in ultra-distance athletes. Recorded intakes may be as low as, or lower than, half the estimated EE [39,40,41].

It is difficult to accurately predict the energetic cost of ultra-distance events and therefore to establish optimal nutritional strategies that both complement the physiological demands of the activity and prevent loss of lean and fat mass is challenging [13]. The EE during the event was not met due to logistical and GI issues attributed to larger, pre-prepared meals. SG experienced motion sickness and felt ill on days 2 and 3 and had to stop the treadmill to ease sickness. As a result, SG relied more on liquid CHO in the form of fruit smoothies and only consumed cold solid foods, such as cereal and sandwiches (Table 1). The initial nutrition strategy would have matched her EE more closely and reduced the observed ED. The very low EI of ~2500 kcal/day over the attempt clearly does not meet the EE of ~ 7000 kcal/day. SG reports only consuming approximately 1000 kcal/day during extreme outdoor events where it is mandatory to carry 2000 kcal/day worth of food. She disposes of the additional food to save weight during these events and therefore this low intake is not unusual for her but should not be considered an ecologically valid nutritional strategy for others engaging in ultra-endurance activity.

SG’s EI throughout the event typifies the EI of ultra-running athletes, where the recorded EI may be ≤50% of estimated EE [12,13,19]. Field research has indicated an hourly range of EIs from 100 to 430 kcal·hr^−1^, and athletes should expect to complete a race with EI of 36% to 54% of EE [30]. Insufficient EI in ultra-running may be attributed to logistical and practical issues but is more often due to suppression of appetite and digestive problems [16,17]. Per hour, SG averaged an EI of 126 to 203 kcal·hr^−1^ (17 to 37 g·CHO·h^−1^) of running, with an EE of 286 kcal·h^−1^ including rest periods and sleep, or 382 kcal·h^− 1^ for active periods only. SG expended 48,147 kcal over the 7 days and her total EI was only 18,918 kcal, resulting in a daily ED of 4175 ± 760 kcal·day^−1^ (29,229 kcal total). Food and fluid intake during a 100 km off road race equated to a similar 176 Kcals per hour, but almost double the fluid intake at 415 mls per hour. However, temperatures were higher during this outdoor event [42]. Savoury foods were also preferred during the latter stages of the race [42], indicating that desired tastes may relate to nutrient availability and requirements [43]. We assume that the ED was compensated for by degradation of subcutaneous adipose tissue and skeletal muscle of the exercising limbs [12].

Initially, SG planned to eat a pre-cooked meal every 3 h during her 30 min break, but she was unable to consume such large quantities in such brief rest periods. Therefore, from the middle of day 2 onward, she relied on CHO dense foods such as sandwiches, cereal, fruit smoothies, fruit, energy drinks, and chocolate, avoiding heavier high fat foods that SG attributed to past GI distress during ultra-running. Fuelling with a combination of carbohydrates in this manner may attenuate GI distress [14,19]. Glucose is absorbed via the sodium dependent glucose SGLT1 receptor and fructose is transported via the sodium independent GLUT5 transporter [44]. Thus, if too much glucose is ingested, it oversaturates the transporters and causes GI distress. A combination of sugars was ingested during the current study, with a maltodextrin-based sports beverage, smoothies, and solid food used to meet CHO intake. SG did not complain of GI distress but did alter her nutrition strategy to include fewer solids and more liquid food; savoury foods were also preferred later in the event, as per Moran et al. [42].

Although CHO oxidation is more aerobically efficient than fat oxidation (5.0 kcal·L^−1^ of O_2_ consumed produced when oxidising 100% CHO compared with 4.7 kcal·L^−1^ of O_2_ from 100% fat [45]), SG primarily oxidised fat throughout the event (RER: 0.8 ± 0.03), most likely due to intra-muscular adaptations driven by high training volumes and regular exposure to ED [46]. The main reasons for fatigue in ultra-running are most likely central fatigue, thermal stress, muscle damage, and/or limited endogenous muscle glycogen content and blood glucose availability [12]. Fatigue may occur in ultra-running due to depletion and insufficient repletion of endogenous muscle glycogen content undefined but can be delayed by either increasing exogenous CHO fuelling and/or increasing endogenous fat oxidation rates at a given intensity. We know that fat is the primary fuel substrate utilised during low intensity exercise [47,48,49] and is further increased with branched chain amino acid oxidation and carbohydrate depletion [50]. Energetic demands of a prolonged exercise bout (>10 h) at a sufficiently low intensity (~45–60% VO_2max_) could theoretically be attained by fat adapted athletes oxidising exclusively fat (700–800 kcal·h^−1^; [51,52,53]). However, in most races, there are tactical scenarios and topographical challenges that benefit from CHO oxidation, so CHO intake is to be encouraged in ultra-running undefined, with CHO ingestion also shown to have anti-nociceptive qualities undefined.

Costa [19,23] and Jeukendrup [22] have outlined the need for athletes to “train their gut” to allow for improved gastric emptying and absorption, as well as reducing the chances and/or severity of GI complaints. Recommendations for CHO intake for highly active athletes are 5–8 g/kg/day, although recommendations for those in high volume intense activity of 3–6 h/day may need to consume as much as 8–12 g/kg/day [16].

Runners are advised to drink approximately 400–800 mL per hour depending upon weight and intensity of running [54]. Some experienced athletes tolerate mild dehydration better, although for longer events, the detrimental effects can become problematic. Fluid intake in the current study averaged 200 mL per hour of running. Although we did not assess sweat loss or urine output, the stable weight throughout the event suggests that the fluid intake was adequate throughout exercise. The importance of sodium for exercise performance is well documented and to avoid muscle cramps and hyponatraemia, it is recommended that athletes should ingest ~500–1000 mg/h as a starting point. This should be practiced in training in order to establish GI tolerance [55,56]. Daily sodium intake averaged 4577 mg per 18 h of running, or 254 mg/h. This shows that SG did not meet the recommended sodium intake recommendations, most likely as she did not consume sufficient sports-specific drinks with added sodium and most of the sodium she consumed came from food.

We did not assess body composition, but data from previous work [33,57] suggests an ED of this magnitude would lead to a loss of ~2 kg of fat or ~4 kg of muscle; most likely a combination of the two. In fact, SG finished the event 1 kg heavier than she started. The increase in weight is most likely due to an increase in total body water (TBW; [11]). It is unlikely that SG experienced oedema due to fluid overload as she only consumed 3.6 L·day^−1^ of fluid, which is a modest ~200 mL·h^−1^, compared with 500–700 mL·h^−1^ in reported cases of oedema [11]. Increases in TBW can be attributed to several different mechanisms: protein catabolism with hypoproteinaemic oedema [58], increased protein synthesis with increased plasma volume [59], increase of plasma volume due to sodium retention because of increased aldosterone [8], or impaired renal function due to skeletal muscle damage [59]. Milledge et al. [10] found that the retention of sodium led to a positive water balance with a shift of fluid from the intracellular to the extracellular space, as well as a significant correlation with an increase of leg volume. This suggests that sodium retention might lead to oedema during prolonged exercise. Although we did not assess leg volume, anecdotally, our athlete suffers from swelling to the legs and feet, often resulting in having to wear shoes two sizes bigger than usual during the latter stages of competitions.

Increases or stability in BW in spite of large ED may also be due to plasma volume expansion, as estimated from changes in haematocrit and haemoglobin [60]. Alterations in SG’s blood parameters over the course of the event (Hb from 14.6 ± 1.3 g·dl to 13.3 ± 1.9 g·dl; HCt from 43 ± 2% to 39 ± 5%) are indicative of a 9.8% increase in blood volume [13,61]. Gastmann et al. [62] previously documented an increase in plasma volume and serum urea, as well as a decrease in haemoglobin and haematocrit, following an ultra-triathlon. These changes could not be explained by haemoconcentration but were related to a suppressed renal function with reduced renal blood flow, decreased glomerular filtration rate, and increased hyperaldosterone-related renal sodium uptake, as well as to proteolysis during prolonged exercise. The latter may potentially explain the changes in haemoglobin and haematocrit in the present study. It is postulated that measurement of creatine kinase would have provided evidence of rhabdomyolysis [63,64].

Pain may contribute to event termination in ultra-running [1], but pain tolerance is reportedly higher in ultra-runners than in other athletic and sedentary populations [4]. SG reported that her feet and back were at times extremely painful, due to blisters and swelling, and bathed her feet in an ice bath during every break to attenuate these factors. Despite continued pain, following her successful attempt, SG remarked that the record was “easy, but tedious”, indicating that she perceived this as a greater mental than physical challenge.

The management of sleep deprivation is also central in prolonged ultra-marathons [65]. SG reported not sleeping at all during the first three nights and commented that she wished she had continued to run. As the attempt progressed, she became more fatigued and reported sleep duration improved. Interestingly, she describes the tiredness not being as severe as that experienced during a 24 h race and only experienced “normal tiredness” in her legs. From the fourth day onwards, SG battled mid-afternoon waves of tiredness and started feeling fatigued earlier each evening, as the monotony of the event became obvious. Overall, her assessment of the attempt was that this was “an easy WR to break” and that she experienced “no big levels of stress”. This exemplifies the necessity of mental resilience and the ability to cope with monotony and highlights the need for an easily administered tool that can reliably quantify the psychological perturbations that are unique to prolonged ultra-running. RPE may neither be sensitive enough nor appropriate to capture event demands, which remained relatively stable throughout SG’s performance and our previous case study [1].

As well as her ability to exercise very economically, SG’s success can also be attributed to very consistent pacing and well-structured breaks. An even pacing strategy seems crucial in ultra-running to preserve exercise capacity. For example, in 24-h ultra-marathoners, the fastest runners start at lower relative intensities and employ a more even pacing strategy than slower runners [66]. Holt et al. [67] previously found that like SG, successful ultra-marathon finishers break the event down into smaller stages, paying attention to running speed, nutrition, and hydration. Following the event, upon questioning, SG explained that she never considered the total distance, only concentrating on each 3-h block, during which she aimed to complete 21–22 km, knowing that if she completed 120 km each day, she would break the record. Her strategy was to continuously calculate her pace and distance during each 3-h block, something that she used to distract herself.

A final consideration is that of age as age may reflect not just training status, but intra-muscular adaptations and tactical experience, especially in ultra-running, as participant age for best performance in ultra-marathon appears to increase with increasing distance and/or duration [68]. Knechtle and Nikolaidis [13] support these findings, with data for performances ranging from 6 h to 10 days’ duration, with the average age for peak performers at 6- and 10-day events being 44.8 years (43.9–45.7) and 44.6 years (42.9–46.3), respectively. SG was 47 years old when she set this WR and it is highly likely that experience and physiological changes related to her continuous training helped her complete this challenge [69]. Even with continuous training, ageing leads to reductions in VO_2max_, velocity at lactate threshold, maximal heart rate, and muscle strength [70]. Although VO_2max_ typically declines by ~10% per decade after the age of 25–30, this rate of decline is ~50% smaller in endurance athletes [71,72]. Sub-maximal exercise economy is preserved if training is continued, in part due to maintenance of muscle fibre composition [70]. The fact that SG has been continuously training has probably led to her high exercise economy and fostered an ability to tolerate high training loads. Typically, the decline in exercise capacity with ageing is related to a decreased training volume, usually due to reduced recovery and motivation [70]. However, SG’s training load has been consistently high and although there most likely will have been declines in VO_2max_ and skeletal muscle characteristics, these will have been compensated for by higher oxidative enzyme activity, enhanced mitochondrial function [70,73], and maintenance of VO_2_ kinetics [74], thus preserving performance or at least suppressing a decline in it.

### 4.1. Limitations

We did not assess body composition, either via sum of skinfolds or bioelectrical impedance, which would have allowed assessment of (compartmental) changes in body composition. However, collection of these data would likely have furthered time constraints and disrupted the athlete as the data collection that took place was already quite time intensive. Due to equipment failure, we only have heart rate data for one 24 h period, thus not allowing us to describe the full HR profile for the 7 days. However, previous calculations of EE have only been estimated via HR, which may be transiently elevated by factors such as heat, anxiety, stimulant use, and dehydration. We are confident that our EE calculations are representative due to our consistent VO_2_ measurements throughout the event.

### 4.2. Conclusions and Practical Recommendations

Running for 7 days on a treadmill and covering a world-record beating 833.05 km led to a substantial ED of 21,158 kcal, but surprisingly no weight loss and instead, a 1 kg weight gain. The weight gain was possibly the result of an increase in TBW, which was likely due to sodium retention, leading to a positive water balance; this is supported by the drop in Hb from 13.7 g/dL to 11 g/dL. The ED was due to the difficulty of consuming sufficient energy-dense foods without GI distress, which led the athlete to rely more heavily on liquid nutrition and smaller, more frequent meals of snack-type food. Despite this large ED, the EI and oxidation of CHO, fat, and lactate were adequate enough to maintain blood glucose for the majority of the event.

Consistent pacing and regular breaks for fueling and recovery are necessary to sustain performance throughout long ultra-running events such as this. This case study demonstrates the difficulty of consuming sufficient calories during ultra-running, even with careful nutritional planning and with the event taking place in a seemingly optimal environment. Future studies of this kind should include bioelectrical impedance analyses to track changes in body water, muscle, and fat mass.

## Figures and Tables

**Figure 1 ijerph-17-05962-f001:**
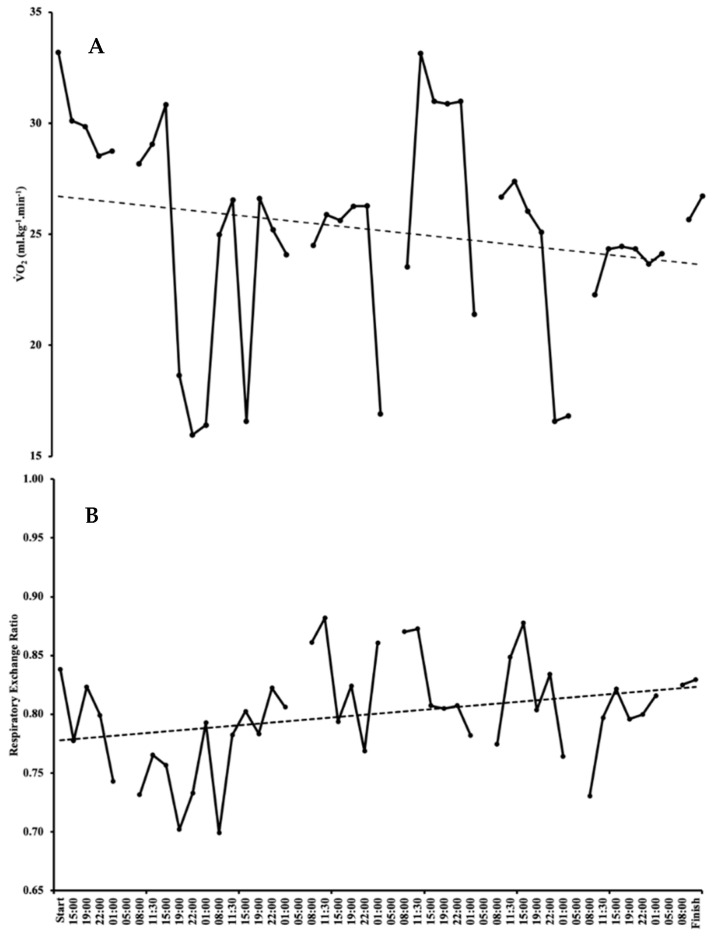
Perturbations in VO_2_ (**A**) and RER (**B**) during the world record attempt, with accompanying trend lines. Please note missing data are from sleep periods 1–5 a.m.

**Figure 2 ijerph-17-05962-f002:**
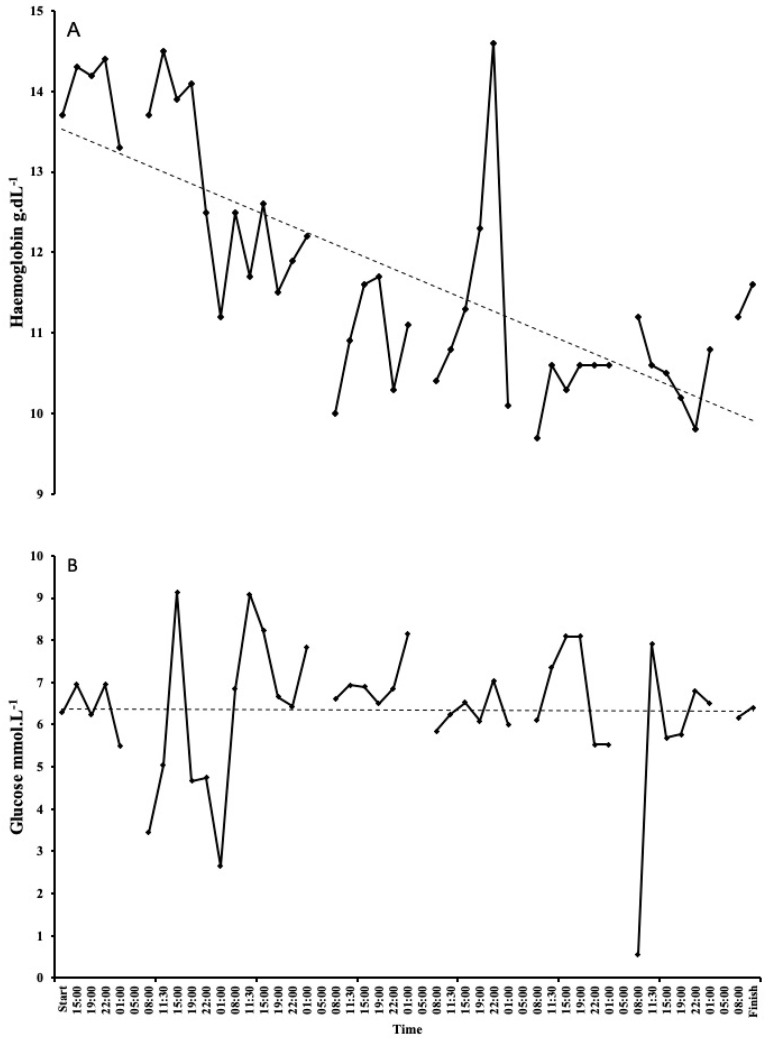
Changes in haemoglobin (**A**) and glucose (**B**) during the world record attempt. Haemoglobin decreased daily, whereas glucose remained remarkably stable. Decreases in glucose were due to periods of gastrointestinal distress, leading to a low intake of food. Please note missing data are from sleep periods 1–5 a.m.

**Figure 3 ijerph-17-05962-f003:**
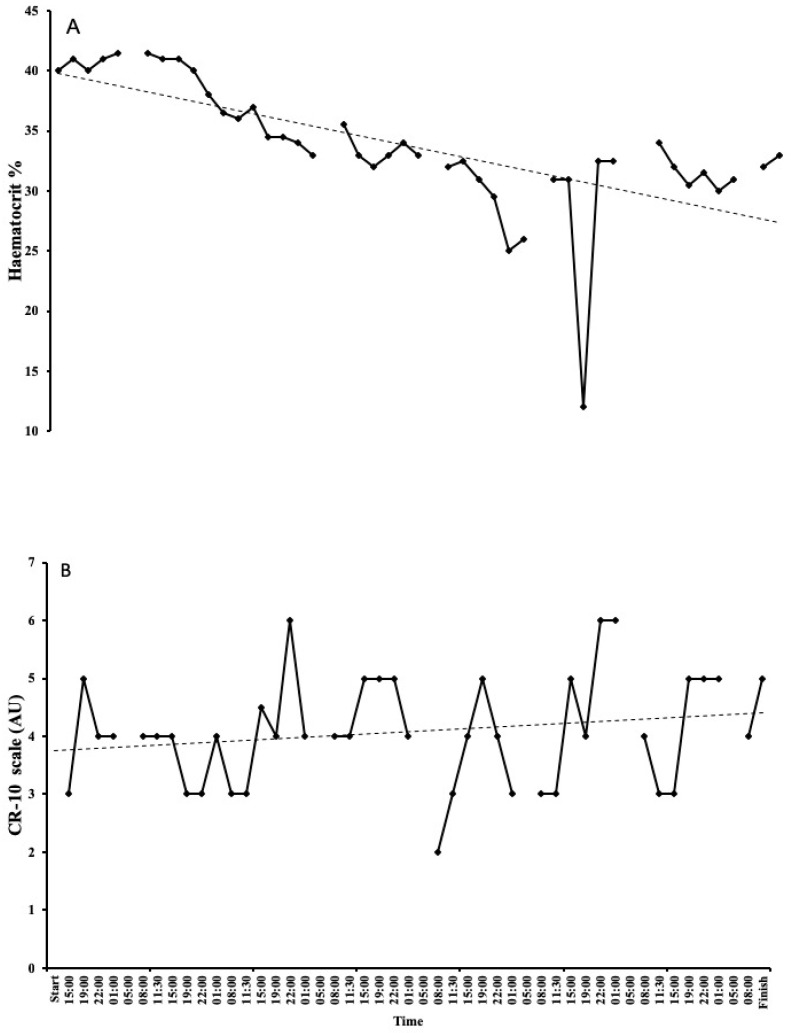
Changes in haematocrit (**A**) and rating of perceived exertion (RPE) (**B**) during the world record attempt. Haematocrit decreased daily. RPE remained stable and higher readings were typically seen later in the day. One very low reading coincided with a period of walking. Please note missing data are from sleep periods lasting from 1–5 a.m.

**Figure 4 ijerph-17-05962-f004:**
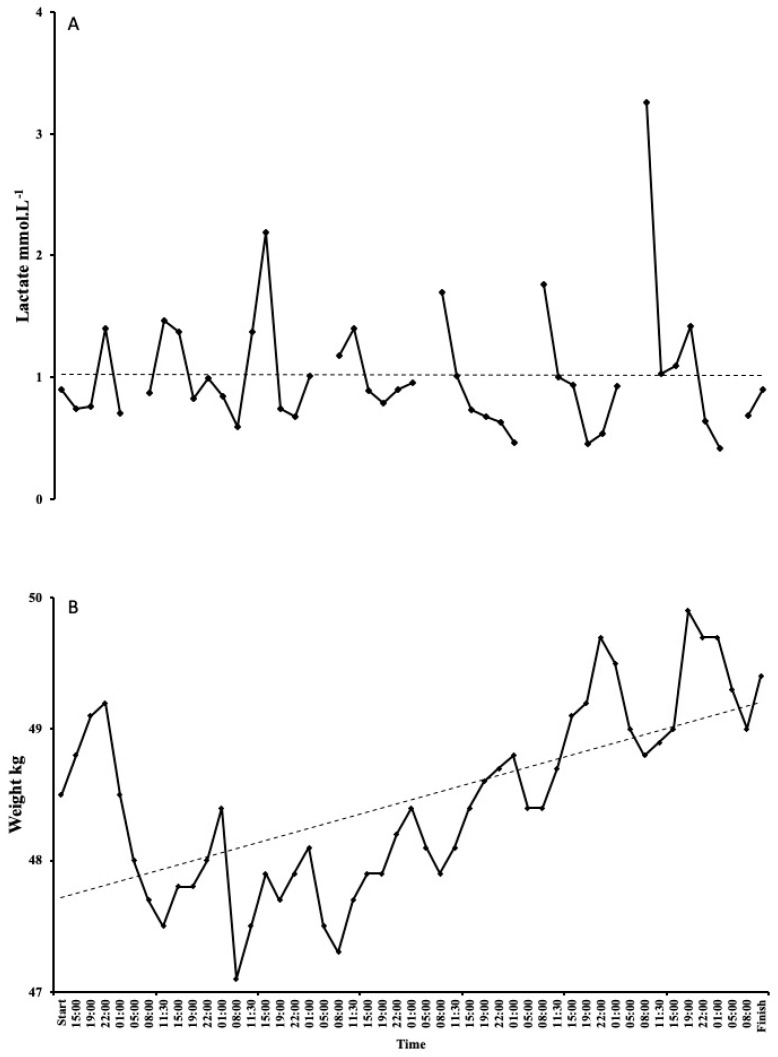
Changes in lactate (**A**) and weight (**B**) during the world record attempt. Lactate remained stable, with only one reading exceeding 3 mmol·L^−1^. Weight initially decreased over the first 3 days, but then continually increased, with SG finishing the event 1 kg heavier than at the start. Please note missing data are from sleep periods 1–5 a.m.

**Table 1 ijerph-17-05962-t001:** Example of food and fluid intake for days 1 and 7 during the world record attempt.

Day 1			Day 7		
Time	Food	Quantity	Time	Food	Quantity
12:20	Carbohydrate Drink	250 mL	04:50	Activia yoghurt	60 g
12:53	Carbohydrate Drink	250 mL	04:50	Tea	50 mL
13:15	Carbohydrate Drink	250 mL	07:10	Mango, passion fruit smoothie	100 mL
13:30	5 grapes		07:10	Strawberry and banana smoothie	100 mL
13:40	Carbohydrate Drink	250 mL	08:00	Ham Sandwich, crisps	165 g
14:05	Carbohydrate Drink	250 mL	08:00	Coffee	300 mL
14:05	5 grapes		08:45	3 minstrels	8 g
14:50	Carbohydrate Drink	250 mL	08:50	3 minstrels	8 g
15:06	Fish pie	300 g	08:55	3 minstrels	8 g
15:06	Coffee	250 mL	09:34	3 minstrels	8 g
15:55	Carbohydrate Drink	250 mL	10:20	Mango, passion fruit smoothie	100 mL
16:20	Carbohydrate Drink	250 mL	10:25	Strawberry and banana smoothie	100 mL
16:40	Carbohydrate Drink	250 mL	11:00	3 minstrels	8 g
17:00	Carbohydrate Drink	250 mL	11:05	Strawberry and banana smoothie	100 mL
17:21	Carbohydrate Drink	250 mL	11:30	Ham Sandwich, crisps	210 g
17:40	Carbohydrate Drink	250 mL	11:30	Coffee	300 mL
18:20	Carbohydrate Drink	250 mL	13:45	Water	250 mL
18:43	Pasta meal with chicken	532 g	13:50	Pineapple, banana and coconut smoothie	100 mL
18:43	Coffee	300 mL	14:30	3 chocolate minstrels	8 g
19:24	Carbohydrate Drink	250 mL	15:00	Cheese sandwich, crisps	
20:11	Carbohydrate Drink	250 mL	16:25	Pineapple, banana, and coconut smoothie	100 mL
21:13	Coke	250 mL	16:55	Water	250 mL
21:13	Water	250 mL	16:58	Strawberry and banana smoothie	100 mL
21:45	Water	250 mL	18:00	Mango, passion fruit smoothie	100 mL
22:16	Rice and mince + water	208 g	18:15	Strawberry and banana smoothie	100 mL
22:16	Coffee	300 mL	18:30	Sandwich	
23:10	Water	250 mL	18:30	Coffee	300 mL
00:25	Water	250 mL	20:00	Strawberry and banana smoothie	100 mL
00:26	Orange juice	200 mL	20:00	Water	250 mL
01:00	Water	250 mL	20:20	Mango, passion fruit smoothie	100 mL
01:18	Tea	300 mL	21:35	Mango, passion fruit smoothie	100 mL
			23:00	Pineapple, banana, and coconut smoothie	100 mL
			23:15	Strawberry and banana smoothie	100 mL
			00:05	Water	250 mL
			00:05	Mango, passion fruit smoothie	100 mL
			00:06	5 Grapes	
			00:20	5 Grapes	

**Table 2 ijerph-17-05962-t002:** Daily energy intake derived from carbohydrate (CHO), protein, and fat during the world record attempt.

Day	Total (kcal)	Energy Intake (kcal)			Absolute and Relative Intakes (g)		
		CHO	Protein	Fat	CHO	Protein	Fat
		(% Total)	(% Total)	(% Total)	(g·kg^−1^)	(g·kg^−1^)	(g·kg^−1^)
1	3661	2672	568	576	668	142	64
		(73%)	(15.5%)	(15.7%)	(13.7)	(2.9)	(1.3)
2	2475	1484	584	495	371	146	55
		(60%)	(23.6%)	(20%)	(7.6)	(3.0)	(1.1)
3	2657	1916	312	441	479	78	49
		(72.1%)	(11.7%)	(16.6%)	(9.9)	(1.6)	(1.0)
4	2901	2288	264	306	572	66	34
		(78.9%)	(9.1%)	(10.5%)	(11.8)	(1.4)	(0.7)
5	2518	1496	312	693	374	78	77
		(59.4%)	(12.4%)	(27.5%)	(7.7)	(1.6)	(1.6)
6	2432	1228	356	846	307	89	94
		(50.5%)	(14.6%)	(34.8%)	(6.3)	(1.8)	(1.9)
7	2274	1268	316	684	317	79	76
		(55.8%)	(13.9%)	(30.1%)	(6.5)	(1.6)	(1.6)
8	716	544	56	108	136	14	12
		(76%)	(7.8%)	(15.1%)	(2.8)	(0.3)	(0.3)
Average *	2703	1765 (65%)	387 (14%)	577 (21%)	441	97	64
Total	19634	12896	2768	4149	3224	692	461

Note: Eight days are recorded due to the start time of the attempt being 12:00 p.m.; * average excludes day 8 as this was not a full day.

**Table 3 ijerph-17-05962-t003:** Daily macronutrient intake for 20 waking hours derived from CHO, protein, and fat, expressed as intake (g) per hour.

Day	CHO	Protein	Fat
	(g·hr^−^^1^)	(g·hr^−^^1^)	(g·hr^−^^1^)
1	33	7	3
2	19	7	3
3	24	4	2
4	29	3	2
5	19	4	4
6	15	4	5
7	16	4	4
8	7	1	1
Average *	22	5	3
SD	6.9	1.6	1.0

Note: Eight days are recorded due to the start time of the attempt being 12:00 p.m.; * average excludes day 8 as this was not a full day.

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
