# Peer review of "Physiological Responses and Nutritional Intake during a 7-Day Treadmill Running World Record"

_ijerph, 2020, doi:10.3390/ijerph17165962_

Round 1

Reviewer 1 Report

The research carried out is very interesting, I think the scientific literature needs more case reports that provide information on specific interventions. However, some considerations must be taken into account to improve said research.

Introduction section:

  • Add aspects related to nutrition, importance of adequate dietary and nutritional intake in training, competitions or specific situations (other case reports).

Materials and Methods section:

  • Delete the athlete's name (initials of the name only) initials and indicate that it is a female athlete.
  • Was there a specific nutritiondal planing? To indicate that it was taken into account to develop it or how it was developed
  • Was any type of nutritional supplement prescribed? (for example, caffeine, sports bar, sports gel, sports drink,...)
  • Why weren't micronutrient and liquid intake calculated? sodium and liquids are important in these situations

Results section:

  • It would be interesting to indicate type food and supplements that athlete have consumed.
  • Is it possible to indicate the nutritional intake per hour? This aspect is considered by current nutritional intake recommendations during competition.
  • Were the nutritional recommendations for the competition met?

Discussion section

  • Discuss about nutritional recomendation (liquid, sodium and carbohydrate), nutritional and supplement prescription, gastric training.
  • Compare the nutrient intake results with other case studies.
  • As it was a research evaluated in 2011, some nutritional recommendations have hanged.

A section of conclusions and/or practical applications may be useful.

Author Response

The research carried out is very interesting, I think the scientific literature needs more case reports that provide information on specific interventions. However, some considerations must be taken into account to improve said research.

Thank you for your detailed feedback and kind comments. Please see below for our responses and the changes that we’ve made.

Introduction section:

  • Add aspects related to nutrition, importance of adequate dietary and nutritional intake in training, competitions or specific situations (other case reports).
  • We agree that this will strengthen the introduction and have added a paragraph to include this.

Materials and Methods section:

  • Delete the athlete's name (initials of the name only) initials and indicate that it is a female athlete.
  • Her name has been deleted and I’ve added that she is female.
  • Was there a specific nutritiondal planing? To indicate that it was taken into account to develop it or how it was developed
  • There was an initial plan. We have now added a paragraph to the methods section explaining this.
  • Was any type of nutritional supplement prescribed? (for example, caffeine, sports bar, sports gel, sports drink,...)
  • SG only consumed a carbohydrate sports drink, but no other supplement. She only had small amounts of caffeine from instant coffee. We have added this information to the methods section.
  • Why weren't micronutrient and liquid intake calculated? sodium and liquids are important in these situations
  • We agree that these are important, and these were calculated and are shown in the text just above table 2. (You might have missed these as they are not shown in a table). Other than sodium we did not add other micronutrient data as we felt it was not relevant to the overall aim of the paper.

Results section:

  • It would be interesting to indicate type food and supplements that athlete have consumed.
  • We agree and we have included a table with examples of nutrition and fluid intake for days 1 and 7 to illustrate the change in what SG consumed.
  • Is it possible to indicate the nutritional intake per hour? This aspect is considered by current nutritional intake recommendations during competition.
  • The data for this is displayed in Table 2, where we show absolute and relative intake for each day and the total. The g/kg is displayed in brackets below the total g. We have now also added the g/hour in Table 3.   
  • Were the nutritional recommendations for the competition met?
  • There was a large energy deficit, and we now expand more upon this in the discussion. The initial plan of consuming several larger meals was abandoned due to logistical and gastro-intestinal issues. We have added commentary for this in the discussion.

Discussion section

  • Discuss about nutritional recomendation (liquid, sodium and carbohydrate), nutritional and supplement prescription, gastric training.
  • Compare the nutrient intake results with other case studies.
  • As it was a research evaluated in 2011, some nutritional recommendations have hanged.

Thank you for pointing these out. We have now added content to address your recommendations.

A section of conclusions and/or practical applications may be useful.

We have now added a conclusion and practical recommendation and agree that the paper finishes on a much more positive note. 

Reviewer 2 Report

The manuscript describes a case study of a 47-year old women’s successful attempt at a 7-day treadmill running world record. The manuscript is well written, and I did not find many grammatical or spelling errors. I think the manuscript presents some interesting and important insights, as reports of extreme physiology are, by definition, rare. Thus I think that fact alone provides some strong novelty to the present report. I have a few consideration for the authors below, but overall I think this was a very good report of the attempt and I thoroughly enjoyed reading it. Well done.

Line 68: I would say how old SG is currently is irrelevant as people could read this document any number years from now. I would just report how old SG was at the time of the attempt.

Line 77 – 82: I would just change the ordering of sentences and put the last sentence of the paragraph as the first. That way the first thing that is mentioned is exactly what world record attempt took place.

Line 118 – 123: This is arguably my biggest gripe with the report. Considering the author’s highlight (e.g., page 1 line 41, page 10 line 251) the potential loss of muscle mass that can occur as a physiological response to ultra-running, how can negligible protein oxidation be assumed? Considering the large energy deficit being reported during this attempt is this really an accurate assumption? If so this would need to be expanded on in the discussion. At the very least the authors should acknowledge that these values are likely overestimated.

Table 1: I would make the “Energy Intake (kcal)” and “Absolute and relative intakes (g)” stand out more. So I’d bold them and not italicize, as of now they get lost in the rest of the figure and thus don’t direct the reader.

Page 163: Add in what day

Results: I like the graphs that the authors have included but considering the nature of the report I would personally like to see the VO2 and RER graphed out. The variability, in my opinion, would be interesting.

Line 203: The authors mention measuring fluid intake, is this really the same thing as “hydration levels?”

Line 213 – 214: All RER can tell us is the proportion of CHO and FAT being utilized so I don’t think we can assume this. I think more support is need for this. Why do your results suggest this statement? Wouldn’t SG utilize endogenous CHOS stores and exogenous fat stores efficiently too?

Line 219: Correct me if I’m wrong but I think this is the first use of the “UM” abbreviation. If so please define it, if not my apologies.

Line 236: Same as above, this is the first time “PV” is used. Prior to this the entire phrase was used.

Line 307 – 313: This is an interesting section, but right now it doesn’t provide any insight as to why SG’s age was useful beyond saying individuals of similar ages do well in ultra running. I don’t think an entire dissertation is required but a brief expansion on the physiological changes that occur at these ages would be pertinent here.

Line 315: I understand timing was an issue, and I agree skinfolds, DEXA, and BodPod probably took up too much time. However, bioelectrical impedance takes all of 30 seconds for the actual test. It’s not a deal break that its not in the manuscript, but I wouldn’t say the reason is because bioelectrical impedance requires a great deal of time. I do appreciate that the authors are including this as a limitation.

Lastly, I as a reader would appreciate a conclusion sentence of some type, ending on the limitations is not a great way to go. What information can be gleaned from this report, why it important, etc. tie a nice bow (so to speak) onto this manuscript for me. Once again great work by the authors.

Author Response

The manuscript describes a case study of a 47-year old women’s successful attempt at a 7-day treadmill running world record. The manuscript is well written, and I did not find many grammatical or spelling errors. I think the manuscript presents some interesting and important insights, as reports of extreme physiology are, by definition, rare. Thus I think that fact alone provides some strong novelty to the present report. I have a few consideration for the authors below, but overall I think this was a very good report of the attempt and I thoroughly enjoyed reading it. Well done.

Thank you very much for kind comments and feedback. We have made the necessary changes and have outlined these below.  

Line 68: I would say how old SG is currently is irrelevant as people could read this document any number years from now. I would just report how old SG was at the time of the attempt.

We agree that it is not relevant. This has now been omitted.

Line 77 – 82: I would just change the ordering of sentences and put the last sentence of the paragraph as the first. That way the first thing that is mentioned is exactly what world record attempt took place.

We have changed this and agree the structure has been improved.

Line 118 – 123: This is arguably my biggest gripe with the report. Considering the author’s highlight (e.g., page 1 line 41, page 10 line 251) the potential loss of muscle mass that can occur as a physiological response to ultra-running, how can negligible protein oxidation be assumed? Considering the large energy deficit being reported during this attempt is this really an accurate assumption? If so this would need to be expanded on in the discussion. At the very least the authors should acknowledge that these values are likely overestimated.

We take on board your point and agree that there must have been some protein oxidation, although its calculation / estimation is difficult. We have now acknowledged that there must have been protein oxidation and as a result that the values are most likely an overestimation. This has been addressed in the methods section.  

Table 1: I would make the “Energy Intake (kcal)” and “Absolute and relative intakes (g)” stand out more. So I’d bold them and not italicize, as of now they get lost in the rest of the figure and thus don’t direct the reader.

These are now displayed in bold and we agree that they stand out more.

Page 163: Add in what day

Day ‘1’ has been added.

Results: I like the graphs that the authors have included but considering the nature of the report I would personally like to see the VO2 and RER graphed out. The variability, in my opinion, would be interesting.

We agree that this is something we should display and have added a figure to show these.

Line 203: The authors mention measuring fluid intake, is this really the same thing as “hydration levels?”

No, we agree that hydration levels refer to something different and have changed this to say ‘fluid intake’.

Line 213 – 214: All RER can tell us is the proportion of CHO and FAT being utilized so I don’t think we can assume this. I think more support is need for this. Why do your results suggest this statement? Wouldn’t SG utilize endogenous CHOS stores and exogenous fat stores efficiently too?

Thank you for pointing this out. This is indeed correct, and we have amended the sentence to be less speculative.

Line 219: Correct me if I’m wrong but I think this is the first use of the “UM” abbreviation. If so please define it, if not my apologies.

Thank you for pointing this out. These have been amended.

Line 236: Same as above, this is the first time “PV” is used. Prior to this the entire phrase was used.

Thank you for pointing this out. These have been amended.

Line 307 – 313: This is an interesting section, but right now it doesn’t provide any insight as to why SG’s age was useful beyond saying individuals of similar ages do well in ultra running. I don’t think an entire dissertation is required but a brief expansion on the physiological changes that occur at these ages would be pertinent here.

Thank you for pointing this out. We have now included some content on the physiological changes that occur from ageing.

Line 315: I understand timing was an issue, and I agree skinfolds, DEXA, and BodPod probably took up too much time. However, bioelectrical impedance takes all of 30 seconds for the actual test. It’s not a deal break that its not in the manuscript, but I wouldn’t say the reason is because bioelectrical impedance requires a great deal of time. I do appreciate that the authors are including this as a limitation.

It is something we wish we had included, and it would have added to the overall results and analysis. In fact, the athlete completed a further WR attempt at our University in 2018 and we made sure we included this measure. This manuscript is currently under preparation.

Lastly, I as a reader would appreciate a conclusion sentence of some type, ending on the limitations is not a great way to go. What information can be gleaned from this report, why it important, etc. tie a nice bow (so to speak) onto this manuscript for me. Once again great work by the authors.

We have now included a conclusion and practical applications sections instead of ending on the limitations, which is better, and more positive way to end this

Round 2

Reviewer 1 Report

The research carried out is very interesting, I think the scientific literature needs more case reports that provide information on specific interventions. The authors have appropriately incorporated the suggestions made so the manuscript has improved a lot. Congratulations!